# Study on Photocatalytic Performance of Ag/TiO_2_ Modified Cement Mortar

**DOI:** 10.3390/ma15114031

**Published:** 2022-06-06

**Authors:** Linsong Wu, Xiaofang Pei, Mengjun Mei, Zhen Li, Shiwei Lu

**Affiliations:** 1School of Urban Construction, Yangtze University, Jingzhou 434023, China; wulinsong12@126.com (L.W.); 18077928900@163.com (X.P.); meimengjun2021@163.com (M.M.); lizhen@yangtzeu.edu.cn (Z.L.); 2Hubei Key Laboratory of Water System Science for Sponge City Construction, Wuhan University, Wuhan 430072, China

**Keywords:** Ag-TiO_2_, cement mortar, photocatalysis, pollutant degradation

## Abstract

In this paper, Ag-TiO_2_ photocatalysts with different Ag contents (1 mol%–5 mol%) were prepared and applied to cement mortar. The photocatalytic performance of Ag-TiO_2_ and photocatalytic cement mortar under UV light and simulated solar light was evaluated. The results showed that Ag loading on the surface of TiO_2_ could reduce its band gap width and increase its absorbance in the visible region, and 2% Ag-TiO_2_ had the highest photocatalytic activity under UV light, the degradation rate of methyl orange (MO) was 95.5% at 30 min, and the first-order reaction constant k was 0.0980 min^−1^, which was 61.7% higher than that of TiO_2_, and 5% Ag-TiO_2_ had the highest photocatalytic activity under solar light, the degradation rate of methylene blue (MB) was 69.8% at 40 min, and the first-order reaction constant k was 0.0294 min^−1^, which was 90.9% higher than that of TiO_2_. The photocatalytic mortar prepared by the spraying method has high photocatalytic performance, The MO degradation rate of sample S2 under UV light was 87.5% after 120 min, MB degradation rate of sample S5 under solar light was 75.4% after 120 min. The photocatalytic reaction conforms to the zero-order reaction kinetics, which was 1.5 times–3.3 times higher than that of the mixed samples and has no effect on the mechanical properties of mortar.

## 1. Introduction

The environmental pollution problems faced by the city restrict the sustainable development of the city [1,2,3]. Industrial waste gas and wastewater, automobile exhaust and runoff pollution endanger the urban environment and the health of the people [4,5]. Photocatalytic cement-based materials are effective in alleviating urban pollution and have been widely concerned by researchers [6,7]. At present, it has been applied to urban roads, building exterior walls and indoor materials [8,9,10,11].

Nano-TiO_2_ is the most widely used photocatalytic material with the advantages of strong redox, cheap, non-toxic, high stability and renewable recycling [12]. Under UV light irradiation, TiO_2_ valence band electrons will transfer to the conduction band, thus forming electron-hole pairs, and carriers diffuse to the TiO_2_ surface to generate hydroxyl radicals (OH•) and superoxide radicals (O_2_•−) with strong oxidizing properties and decompose pollutants by redox reaction on the TiO_2_ surface [13]. However, due to its wide band gap (3.2 eV), TiO_2_ can only be activated in ultraviolet light which accounts for 5% of the solar spectrum and cannot use visible light (45%). In addition, photogenerated electrons and holes have short lifetimes and high recombination rates, leading to low quantum efficiency, which all affect the photocatalytic performance of TiO_2_ [14,15]. Researchers have proposed several modification methods to improve the photocatalytic performance of TiO_2_, including:
(1)Doping metal ions or nonmetal ions in TiO_2_ changes its photoelectric characteristics, Metal ions such as Fe^3+^, V^4+^, Cr^3+^ and In^3+^ and non-metal ions such as N, S, C and F can be doped into the TiO_2_ lattice to expand the light absorption range of nano- TiO_2_ from the ultraviolet band to the visible band and improve the photocatalytic and self-cleaning performance under visible light [16,17,18,19]. Pérez-Nicolás et al. [20] used TiO_2_, Fe^3+^-TiO_2_ and V^4+^-TiO_2_ as photocatalysts to prepare photocatalytic mortars and investigated their performance in NO removal under three irradiation conditions (UV, solar light, and visible light). The results show that the photocatalytic activity of Fe- TiO_2_ and V- TiO_2_ mortars were higher than that of TiO_2_ mortars under visible light, but the photocatalytic activity of Fe-TiO_2_ and V-TiO_2_ mortars was lower than that of TiO_2_ mortars in ultraviolet and solar light. Janus et al. [21] evaluated the photocatalytic activity of N, C-TiO_2_ photocatalyst. The results show that the cement board containing N, C-TiO_2_ photocatalyst has higher photocatalytic efficiency than the cement board with TiO_2_.(2)Coupling TiO_2_ with other low-bandgap semiconductors such as ZnO, WO_3_, SnO_2_, CdS, and Fe_2_O_3_ to form heterojunctions can also improve its photocatalytic performance [22,23,24,25]. Wu et al. [26,27] prepared SnO_2_/TiO_2_ and Fe_2_O_3_/TiO_2_ by gaseous detonation method, and the prepared SnO_2_/TiO_2_ and Fe_2_O_3_/TiO_2_ showed higher visible light absorption and photocatalytic efficiency than TiO_2_. Feng et al. [28] prepared TiO_2_/g-C_3_N_4_ modified cement paste and investigated its photocatalytic degradation performance. TiO_2_/g-C_3_N_4_ has a wide light absorption range and low photoinduced carrier recombination rate, degraded rhodamine B (RhB) dye by 97% after 40 min of irradiation.(3)Loading noble metals such as Ag, Au, and Pt on the surface of TiO_2_ can reduce the carrier recombination rate and improve the efficiency of TiO_2_ photocatalysis [29,30,31].


Ag loading on TiO_2_ is one of the successful methods to improve the photocatalytic performance of TiO_2_. Ag on the TiO_2_ surface can act as an electron trap to trap the electrons transferred from the TiO_2_ conduction band; Meanwhile, it creates a surface plasmon resonance (SPR) effect to extend the light absorption to the visible region; In addition, Ag nanoparticles have this excellent antibacterial property [32,33]. Ling et al. [34] reported the preparation of Ag/TiO_2_ composites using the photoreduction method for photocatalytic degradation of different organic pollutants and found that Ag could improve the photocatalytic degradation rate, promote the charge separation process, and enhance the adsorption of organic pollutants on the TiO_2_ surface. Shan et al. [35] successfully prepared a set of biochar-coupled Ag-TiO_2_ materials, and the results showed that the photocatalytic degradation performance of Ag-modified TiO_2_ was superior to that of pure TiO_2_ due to the synergistic effect of Ag, TiO_2_ and biochar. The application of Ag-TiO_2_ to cement-based materials can bring pollution removal, self-cleaning and antibacterial properties.

At present, researchers mainly focus on the photocatalytic performance of Ag-TiO_2_, but there are few studies on the application of Ag-TiO_2_ in cement mortar, and the research on the photocatalytic performance of Ag-TiO_2_ in cement mortar was not sufficient. Graziani et al. [36] applied Ag-TiO_2_ to brick samples and tested its inhibition effect on algae. The results confirmed that porosity and roughness play a key role in algal adhesion but did not find that Ag-TiO_2_ significantly improved the algal adhesion of samples. Ren et al. [37] synthesized bamboo charcoal loaded with Ag-doped TiO_2_ (Ag/TiO_2_-BC) and used it in cement-based slurry. Ag/TiO_2_-BC composite material has good hygroscopic and formaldehyde removal performance and has broad application prospects in improving indoor air environments. Yang et al. [38] loaded Ag on the surface of TiO_2_/Zeolite Fly Ash Bead (ZFAB) and found that the content of Ag particles in TiO_2_/ZFAB modified cementing material affected its photocatalytic performance. However, the photocatalytic enhancement of Ag-TiO_2_ on cement-based materials was not evaluated separately, it is necessary to study the photocatalytic properties of Ag-TiO_2_ in cement mortar.

Dye wastewater is one of the main causes of water pollution. The low degradation of this wastewater will cause pollution of surface water and groundwater, and adversely affect the health of animals and human beings. Therefore, it is crucial to remove dyes from water pollution [39,40,41,42,43]. Photocatalytic treatment of water pollution is one of the research hotspots. Mohammad et al. [44,45,46,47] prepared SnO_2_/TiO_2_, Ag@SnO_2_/g-C_3_N_4_, S doped g-C_3_N_4_ and SnO_2_/CeO_2_ nanomaterials and used them for the degradation of water pollutants such as methylene blue (MB), Congo red (CR) and Rhodamine B (RhB) and antibiotic tetracycline (TC) with high degradation rates. Cement mortar is one of the most versatile building materials used in construction projects and is widely used in building walls, ground, roads, and parking lots. Photocatalytic cement mortar is an ideal carrier for treating water pollution. The application of Ag-TiO_2_ to cement mortar has the potential to remove water pollution.

In this study, Ag-TiO_2_ samples with different Ag contents were prepared, its microstructure and optical characteristics were investigated, the cement mortar was prepared by direct incorporation method and spraying method, respectively, and the effects of the two methods on the mechanical properties of the mortar were investigated. The photocatalytic performance of nano-powder and photocatalytic cement mortar was evaluated by using a methyl orange degradation test under UV light and a methylene blue degradation test under simulated solar light.

## 2. Experimental

### 2.1. Materials

AgNO_3_ (AR, Sinopharm Chemical Reagent Co., Ltd., Shanghai, China.), P25 TiO_2_ (Evonik Industries AG, Essen, Germany), Methyl orange (AR, Sinopharm Chemical Reagent Co., Ltd., Shanghai, China.), Methylene blue (AR, Sinopharm Chemical Reagent Co., Ltd., Shanghai, China.), 42.5R Portland cement (Huaxin Cement Co., Ltd., Huangshi, China), Natural Sand, Deionized water and Normal Tap Water were used in this study.

### 2.2. Preparation of Ag-TiO_2_

Silver nitrate solution was prepared by weighing different masses of silver nitrate, and then a certain amount of P25 powder was added. The Ag: Ti mole ratios were 1%, 2% and 5%, respectively, and the nano-TiO_2_ was dispersed in silver nitrate solution by ultrasound for 30 min in a dark room, and then reacted under the irradiation of a 250 W mercury lamp for 2 h. After filtration, cleaning and drying, Ag-TiO_2_ powder was obtained. The prepared sample was shown in Figure 1a.

### 2.3. Preparation of Photocatalytic Mortar

Photocatalytic mortar with Ø 90×20 mm was prepared at the mass ratio of cement: sand: water = 1:2:0.5. For comparison purposes, two kinds of photocatalytic cement mortars were prepared: (1) 5% cement weight of different photocatalysts was ultrasonically dispersed in water for 20 min, added directly to the cement mortar mixture and stirred for 2 min, Nano TiO_2_ powder was directly and closely mixed with the cementing material, after 1 day, the samples were removed from the moulds and put into the standard curing room (25 °C and Relative Humidity > 95%) for 28 days until testing. The samples were designated as M0, M1, M2 and M5, respectively, and the photocatalyst content in each sample was about 2.17 g. (2) The suspension was prepared with different Ag-TiO_2_ and deionized water, and the concentration of the suspension was 20 g/L. After the blank cement mortar sample was demoulded, 20 mL suspension was sprayed on the surface of the sample three times, and the curing method was the same as that of the mixing sample. The samples were designated as S0, S1, S2 and S5, respectively, and the photocatalyst content in each sample was about 0.4 g. The prepared sample was shown in Figure 1b. Cube test blocks (100 mm × 100 mm × 100 mm) were also prepared for compressive strength.

### 2.4. Photocatalysis

The photocatalytic properties of Ag-TiO_2_ samples under UV light were evaluated by the methyl orange degradation test, and those under solar light were measured by the methylene blue degradation test. 250 W mercury lamp as UV light source and 300 W xenon lamp as simulated solar light sources. The initial concentration of MO and MB solution was 10 mg/L, 100 mg Ag-TiO_2_ powder was added to 400 mL MO solution and stirred in a dark room for 30 min to achieve adsorption-desorption equilibrium. After that, illumination was started, and samples were taken every 10 min. After centrifugation, the absorbance of the solution was tested by a 721-spectrophotometer (Shanghai Yoke Instrument Co., Ltd., Shanghai, China).

The photocatalytic performance of cement mortar was also determined by the MO degradation test and MB degradation test. Firstly, put the sample into a beaker, add 400 mL of MO (MB) solution and set it in a dark room for 30 min to achieve adsorption-desorption equilibrium. Second, turn on the illumination for photocatalytic reaction, and solution samples were collected every other 15 min. Finally, the absorbance of the sample solution was measured by 721 spectrophotometry.

### 2.5. Compressive Strength

Control mortar, M2 mortar and S2 mortar with 100 mm × 100 mm × 100 mm were prepared, the specimens were cured for 7 and 28 days, and the compressive strength was tested by a universal testing machine (Hongshan Testing Machine Co., Ltd., Tianshui, China) according to GB/T50081-2019, the loading speed of 0.5 MPa/s was applied during the test. For each different mortar, 6 specimens were prepared for testing (3 test blocks were tested for each age).

### 2.6. Characterization

Philps-PW3040/60 X-ray powder diffractometer (Cu, Kα, λ = 0.15418 nm) with 40 kV tube voltage, 30A tube current, 10–90° scanning angle, and 8°/min scanning speed and FEI-Quanta450 Scanning electron microscopy (SEM) were used to analyze the structure and morphology of the samples. UV–vis (UV-550 UV spectrophotometer, JASCO Corporation, Tokyo, Japan) was used to analyze the optical properties of the samples.

## 3. Results and Discussion

### 3.1. XRD Analysis of Ag-TiO_2_

Figure 2 is the XRD pattern of the sample. The diffraction peaks 2θ = 25.3, 36.9, 38.6, 48.1, 53.9, and 55.1° in the diffraction pattern correspond to the Anatase phases of TiO_2_ (JCPDS No. 21-1272). Other diffraction peaks 2θ = 27.4, 41.2, 54.3 and 56.6° correspond to the Rutile phase (JCPDS No. 21-1276). The intensity of the diffraction peaks of Ag-TiO_2_ is slightly reduced, indicating that the presence of Ag ions reduces its crystallinity, and the angles of the diffraction peaks of Ag-TiO_2_ are not changed, which proves that Ag does not enter the TiO_2_ lattice, which is consistent with the previous conclusions [31]. The samples 2% Ag-TiO_2_ and 5% Ag-TiO_2_ have AgO peaks at diffraction peak 2θ = 32.3°, which proves that a small amount of AgO particles exist on the surface of TiO_2_ from the oxidation of AgO in air. Bian et al. [48] demonstrated that the heterojunction structure of AgO and TiO_2_ is crucial in photocatalytic reactions, which can improve the visible light absorption efficiency of the system and delay the recombination of charge carriers. Similar conclusions were also obtained by Kulal et al. [49]. The diffraction peak of Ag at 2θ = 36.9° was not found in each sample, which was attributed to the coincidence of the diffraction peak with the anatase phase at 2θ = 36.9° or the low content of Ag. The average grain size of all the samples was calculated by Scherrer Formula, and the average grain size of all the samples ranged from 21–22 nm.

### 3.2. SEM and EDS Analysis of Ag-TiO_2_

Figure 3 is the SEM image of the prepared samples. It can be seen from the figure that the morphology of all samples is basically the same, and the particle distribution is relatively uniform, which is spherical particle size between 20–50 nm, consistent with the grain size calculated by the Scherrer formula. There are dendritic agglomerations between the particles, and it is obvious that there are macropore structures formed by particle aggregation connection and mesopores formed by particle accumulation. This indicates that the morphology of nano-TiO_2_ does not change after Ag was loaded, Shokri et al. [50] also observed that the photo-deposition method has no effect on the structure and morphology of TiO_2_, and Ag nanoparticles are located on the surface of TiO_2_ nanoparticles, making it more susceptible to light and improving its photocatalytic activity

Figure 4 shows the Elemental mapping and EDS analysis of the Ag-TiO_2_ sample. The results show that Ag is deposited on the surface of nano TiO_2_ in a point-like distribution. The distribution is not uniform, which may negatively affect its photocatalytic performance. EDS analysis showed that the atomic ratios of Ag/Ti of 1% Ag-TiO_2_, 2% Ag-TiO_2_ and 5% Ag-TiO_2_ were 0.4%, 1.2% and 1.6%, respectively, which were significantly different from the design values, due to the AgNO_3_ solution did not react completely in the reaction.

### 3.3. UV–Vis Spectral Analysis of Ag-TiO_2_

Figure 5 shows the UV-Vis absorption spectra of the samples, and all samples have obvious absorption peaks in the UV region. Compared with pure TiO_2_, the 5% Ag-TiO_2_ absorbance spectrum has a significant red shift. Meanwhile, the absorbance of 5% Ag-TiO_2_ samples is higher than pure TiO_2_ in the visible region. This contributes to the improvement of photocatalytic activity of the samples under solar light.

With *hv* as the horizontal coordinate and (A*hv*)^2^ as the vertical coordinate for the graph, the Eg of each sample can be obtained after making a tangent line to the horizontal coordinate, and the Eg of each sample is 3.2 eV and 3.08 eV, respectively, which shows that Ag loading on the TiO_2_ surface can effectively reduce the forbidden bandwidth of the samples and extend their photo-response range. Furube et al. [51] pointed out that Ag loading on the TiO_2_ surface created a surface plasmon resonance (SPR) effect that extends light absorption into the visible region while increasing the photocatalysis efficiency of TiO_2_ under visible light.

### 3.4. Photocatalytic Analysis of Ag-TiO_2_

Figure 6a shows the degradation rate and photocatalytic rate of methyl orange solution under UV light. The degradation rates of each sample at 30 min were 86.7%, 90.5%, 95.5% and 88.7%, respectively, and all samples were completely degraded at 40 min. Compared with pure TiO_2_, different Ag-TiO_2_ can effectively improve the photocatalytic activity of the sample. Noreen et al. [52] showed that doping of Ag nanoparticles in TiO_2_ is one of the successful methods to inhibit carrier recombination and improve photocatalytic activity. Din et al. [53] pointed out that Ag loading on the surface of TiO_2_ can form Schottky barriers, and Ag acts as an electron trap to capture photo generated electrons and transfer them to oxygen to form superoxide radicals, thus promoting interfacial charge transfer, delaying the recombination of the carrier, and improving the photocatalytic activity of TiO_2_ under UV light. Mogal et al. [54] have also investigated the effect of Ag content on the photocatalytic performance of TiO_2_ and concluded that doping of 0.75 at% had the highest photocatalytic activity under UV light. Sun et al. [55] also found that the photocatalytic performance increased significantly with increasing Ag content until an optimal Ag content with the highest photocatalytic performance. After that, with the increase of Ag content, Ag particles would become carrier recombination centres, resulting in the decrease of TiO_2_ photocatalytic activity.

Figure 6b shows that the photocatalytic degradation of methyl orange solution basically conforms to the first-order reaction equation. The formula is shown below:$$\ln(C_{t}/C_{0})=-k^{1}t$$
where *C*_0_ is the initial concentration (mg/L), *C_t_* is the current concentration (mg/L), *t* is the irradiation time (min), and *k*^1^ is the apparent reaction rate constant (min^−1^).

The apparent reaction rate constants *k*^1^ values of 0.0606, 0.0752, 0.0980 and 0.0628 min^−1^ can be obtained after fitting. The calculated results showed that the photocatalytic rate of the Ag-TiO_2_ samples was significantly higher than that of pure TiO_2_, and the photocatalytic rates of 1% Ag-TiO_2_, 2% Ag-TiO_2_ and 5% Ag-TiO_2_ were increased by 24.1%, 61.7% and 3.6%, respectively, compared with pure TiO_2_. With the increase of the content of Ag, the photocatalytic activity of the sample gradually improves and reaches the highest photocatalytic activity with 2% Ag-TiO_2_. After that, the photocatalytic activity gradually decreases with the increase of the content of Ag. Madhavi et al. and Hou et al. [53] also prepared Ag/TiO_2_ to degrade MO, and the photocatalytic reaction rate was 0.0031 and 0.0018, because of the different preparation methods and reaction conditions.

Figure 7a shows the degradation rate of methylene blue under simulated solar light. The degradation rates of each sample at 40 min were 46.4%, 53.0%, 67.8% and 69.8%, respectively. As shown in the figure, the degradation rate of methylene blue increases with the increase of Ag content in the sample. Komaraiah et al. [56] also evaluated the performance of Ag-TiO_2_ in the degradation of MO and MB under visible light irradiation, the increase of Ag content can enhance the photocatalytic activity of TiO_2_ in the visible region, the results showed that 5% Ag-doped TiO_2_ showed high degradation rate for methyl orange and methylene blue.

Figure 7b shows the fitting line of the degradation rate of methylene blue under solar light, and the results show that the photocatalytic reaction rate conforms to the first-order kinetic equation. The apparent reaction rate constants *k*^1^ values of 0.0154, 0.0184, 0.0275 and 0.0294 min^−1^ can be obtained after fitting. The calculated results showed that the photocatalytic rate of the Ag-TiO_2_ samples was significantly higher than that of pure TiO_2_ under the solar light, and the photocatalytic rates of 1% Ag-TiO_2_, 2% Ag-TiO_2_ and 5% Ag-TiO_2_ were increased by 19.5%, 78.6% and 90.9%, respectively. The higher photocatalytic rate of Ag-TiO_2_ under solar light is because Ag-TiO_2_ has a higher visible light absorption capacity than pure TiO_2_, thus improving its photocatalytic activity. 5% Ag-TiO_2_ has higher photocatalytic activity than 2% Ag-TiO_2_ under solar light_,_ attributed to the fact that loading more Ag will increase the absorbance of TiO_2_ in the visible region and reduce the forbidden band width of TiO_2_, as shown in Figure 5. The same conclusion was reported by other researchers, Harikishore et al. [57] proved that 5 mol% of Ag can reduce the forbidden band width of TiO_2_ composites from 3.2 eV to 2.9 eV, redshift the absorption spectrum to the visible region. Hariharan et al. [58] demonstrated that the response of TiO_2_ nanoparticles to visible light was enhanced after doping with 0.01 M of Ag, resulting in improved photocatalytic performance. Andrade et al. and Muñoz-Fernandez et al. [53] prepared Ag/ZnO for MB degradation, and the photocatalytic reaction rate was 0.007 and 0.017, which proves that Ag-TiO_2_ has high photocatalytic performance.

### 3.5. Photocatalytic Analysis of Cement Mortar

Figure 8 depicts the relationship between the degradation rate of methyl orange and photocatalytic rate and the time of photocatalytic mortar sample under UV irradiation. After 120 min of UV light irradiation, the methyl orange degradation rates of the mixed samples were 25.2%, 29.6%, 30%, and 20.3%, respectively, and the methyl orange degradation rates of the sprayed samples were 83.4%, 84.6%, 87.5% and 73.5%, respectively. Figure 8 also shows that the photocatalytic reaction rate conforms to the zero-order kinetic equation. The formula is shown below:
$$k^{0}t=(C_{0}-C_{t})/C_{0}$$

The apparent reaction rate constants *k*^0^ values were shown in Figure 8. The degradation rate and photocatalytic reaction rate sequence of the samples was S2 > S0 > S1 > S5 > M2 > M0 > M1 > M5. Sprayed samples have a higher degradation rate and photocatalytic rate than mixed samples. The study by Kim et al. [59] also shows that the direct incorporation method is less efficient because most of the TiO_2_ is inside the cement mortar, which cannot be exposed to light and pollutants, and cannot participate in the photocatalytic reaction, resulting in lower utilization of TiO_2_. Meanwhile, the high ionic concentration and high pH environment of cement mortar also have a significant impact on photocatalytic properties [60], and Gupta et al. [61] found that the absorption of light or the competitive capture of redox substances by Ca^2+^, Fe^2+^ and Cl^−^ ions affected the photocatalytic activity of Ag-TiO_2_. Furthermore, MO was negatively charged in solution and repelled by the cement mortar surface, which also leads to the loss of photocatalytic performance [62].

Figure 9 depicts the relationship between the degradation rate of MB and photocatalytic rate and time of photocatalytic mortar sample under solar light irradiation. After 120 min of solar light irradiation, the MB photolysis rate of the blank sample was 16%, and the MB degradation rates of the mixed samples were 30.5%, 33.1%, 33.1%, and 36.7%, respectively, and the MB degradation rates of the sprayed samples were 61.0%, 63.6%, 73.4% and 75.4%, respectively. Figure 9 also shows that the photocatalytic reaction rate conforms to the zero-order kinetic equation. The degradation rate and photocatalytic reaction rate sequence of the samples was S5 > S2 > S1 > S0 > M5 > M2 > M1 > M0. The MB degradation rate and photocatalytic activity of the mixed samples were low under solar light, and it can be considered that there was little difference between the samples. The sprayed samples had a higher MB degradation rate and photocatalytic activity under solar light, and the MB degradation rate and photocatalytic activity increased with the increase of Ag content, which is basically consistent with the aforementioned Ag-TiO_2_ degradation of MB, due to the fact that the spraying method exposes more TiO_2_ to solar light and participates in the oxidation of pollutants in water [5]. Meanwhile, the MB molecules are positively charged in the solution, MB molecule has a positive charge in the solution and is easily adsorbed by TiO_2_ on the surface of cement mortar [62], which also improves the degradation of the MB solution.

### 3.6. Compressive Strength of Cement Mortar

Figure 10 shows the compressive strength of samples Control, M2 and S2 at 7 and 28 days, the results show that the compressive strength at 7 days was 25.3 MPa, 22.4 MPa and 25.9 MPa respectively, and the compressive strength at 28 days was 30.2 MPa, 28.1 MPa and 31.2 MPa respectively. The spraying method basically has no effect on the compressive strength of cement mortar. However, adding 5 wt% Ag-TiO_2_ into cement mortar will reduce its mechanical properties and its 28-day compressive strength will decrease by 7%. Ren et al. [63] reported the same conclusion, which was attributed to the agglomeration phenomenon of excessive TiO_2_ in cement will be more aggravated, leading to the increase of internal pores and defects in cement mortar, reducing its mechanical properties.

## 4. Conclusions

In this study, Ag-TiO_2_ with different Ag contents was prepared by the photodeposition method, and the photocatalytic cement mortar was prepared by the direct incorporation method and the spraying method, respectively. The photocatalytic performance of the samples was evaluated by photodegradation of MB and MO, the main conclusions are as follows:(1)Loading Ag on TiO_2_ did not change the microscopic morphology and crystal structure of TiO_2_, but it could improve the absorbance of TiO_2_ in the visible region and reduce the forbidden bandwidth.(2)Ag-TiO_2_ has high photocatalytic activity under UV and solar light. 2% Ag-TiO_2_ has the highest photocatalytic activity under UV, with a degradation rate of methylene orange of 95.5% in 30 min and photocatalytic reaction rate 61.7% higher than that of TiO_2_; 5% Ag-TiO_2_ has the highest photocatalytic activity under solar light, with a degradation rate of methylene blue of 69.8% in 40 min and photocatalytic reaction rate 90.9% higher than that of TiO_2_.(3)The photocatalytic cement mortar prepared by the spraying method has high photocatalytic activity under UV and solar light, and its photocatalytic reaction rate was 1.5 times–3.3 times higher than that of the mixed sample, and the photocatalyst utilization rate is significantly higher than the mixed sample.(4)Adding Ag-TiO_2_ into cement mortar has a negative effect on its compressive strength, which was reduced by 9% at 28 days, while the spraying method has no effect on the mechanical properties of cement mortar.

## Figures and Tables

**Figure 1 materials-15-04031-f001:**
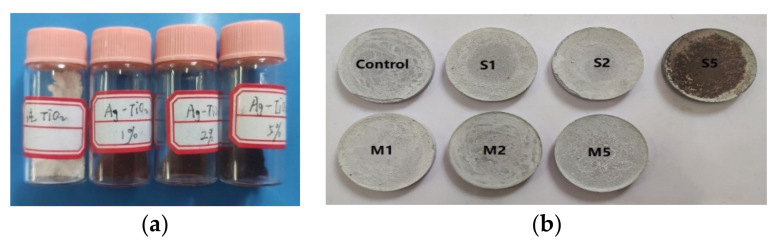
Nano Ag-TiO_2_ powder sample (**a**) and Photocatalytic mortar test block (**b**).

**Figure 2 materials-15-04031-f002:**
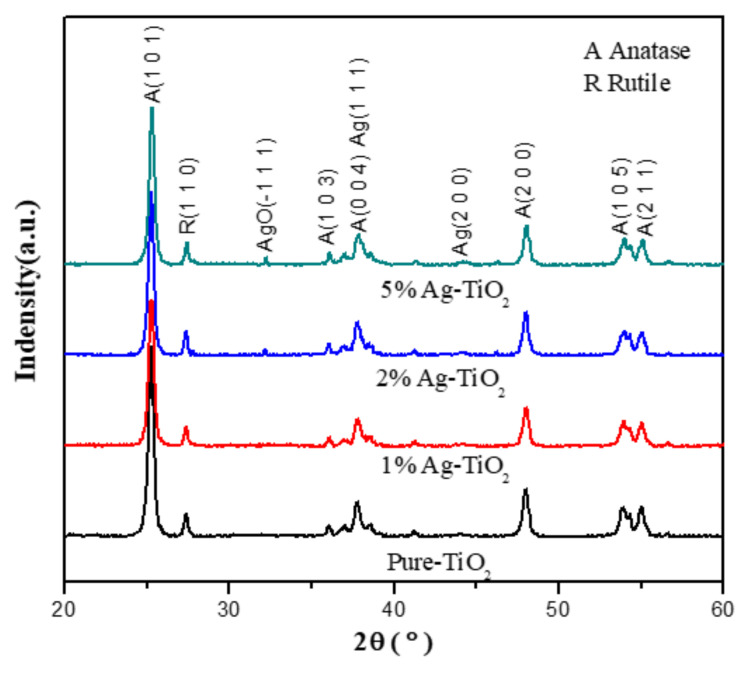
XRD patterns of the sample.

**Figure 3 materials-15-04031-f003:**
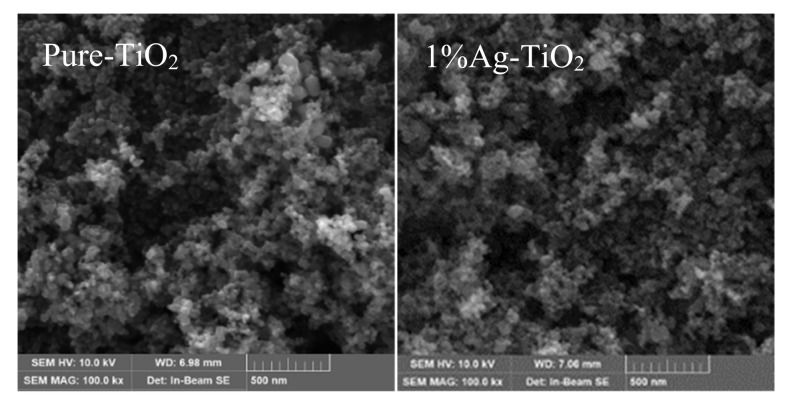
SEM images of Ag-TiO_2_.

**Figure 4 materials-15-04031-f004:**
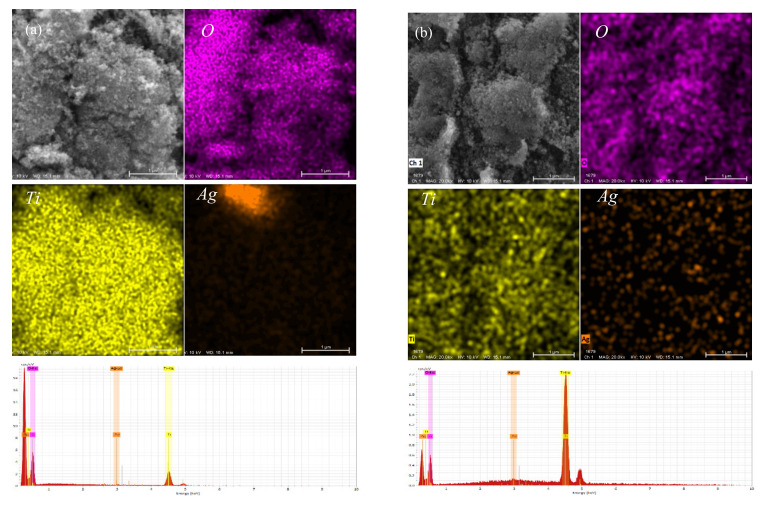
Elemental mapping and EDS analysis of Ag-TiO_2._ (**a**) 1%Ag-TiO_2_. (**b**) 2%Ag-TiO_2_. (**c**) 5%Ag-TiO_2_).

**Figure 5 materials-15-04031-f005:**
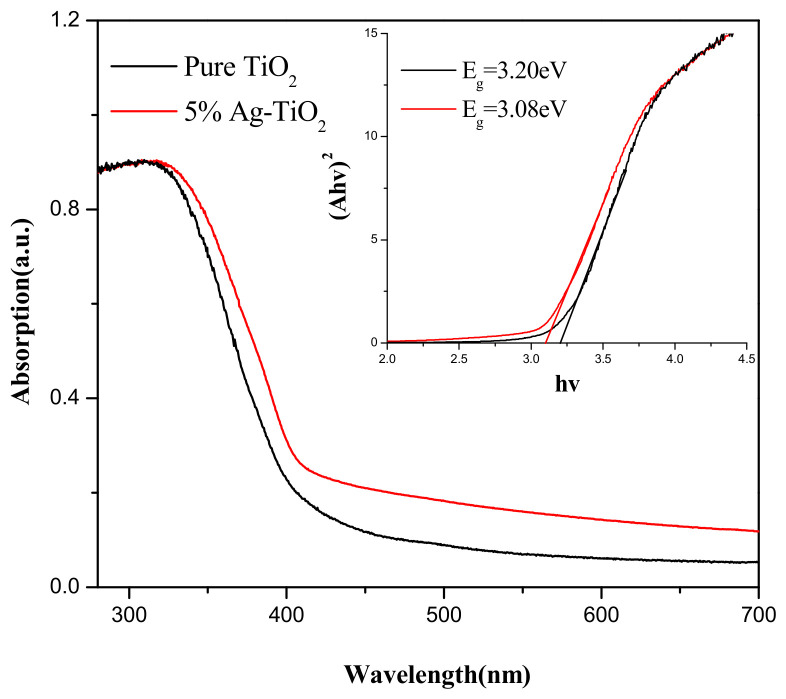
UV-Vis absorption spectra of TiO_2_ and 5% Ag-TiO_2_.

**Figure 6 materials-15-04031-f006:**
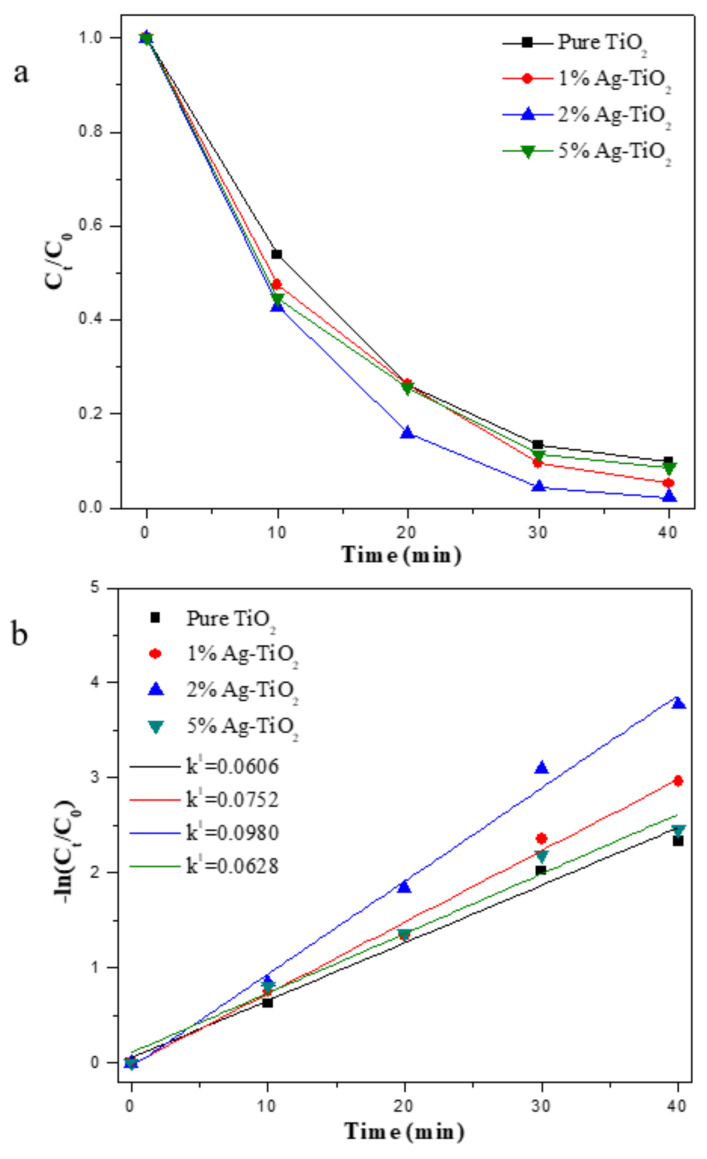
Degradation rate of MO (**a**) and photocatalytic reaction rate (**b**) of TiO_2_ and Ag-TiO_2_ samples under UV light.

**Figure 7 materials-15-04031-f007:**
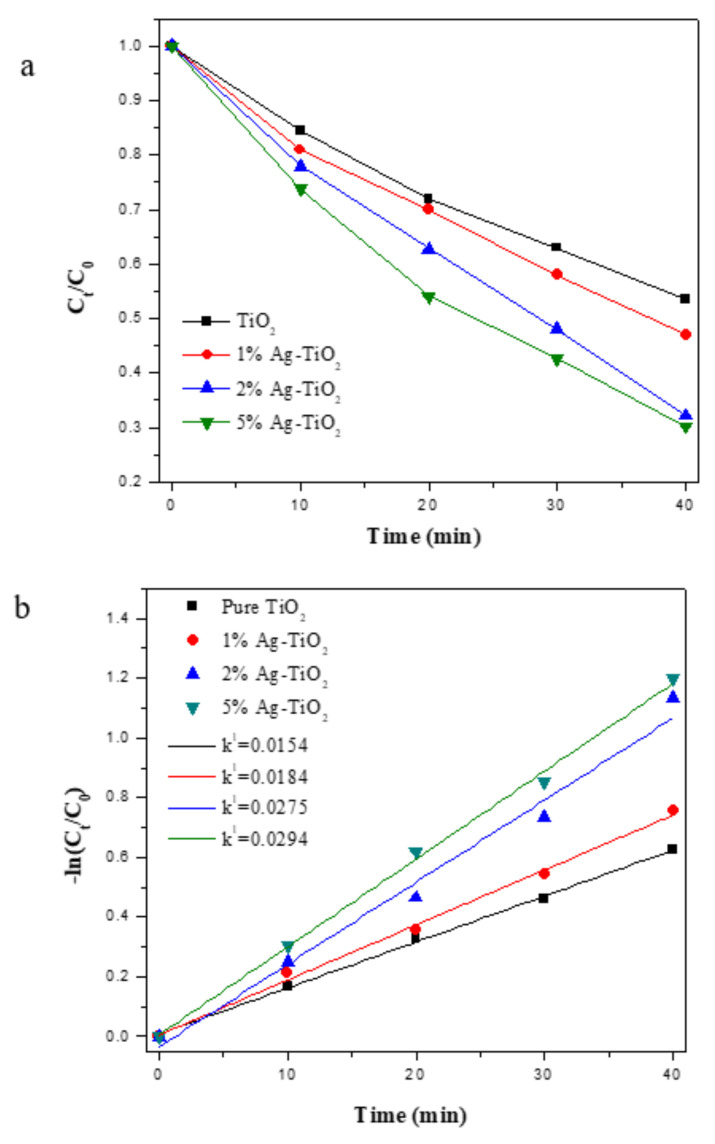
Degradation rate of MB (**a**) and photocatalytic reaction rate (**b**) of TiO_2_ and Ag-TiO_2_ samples under solar light.

**Figure 8 materials-15-04031-f008:**
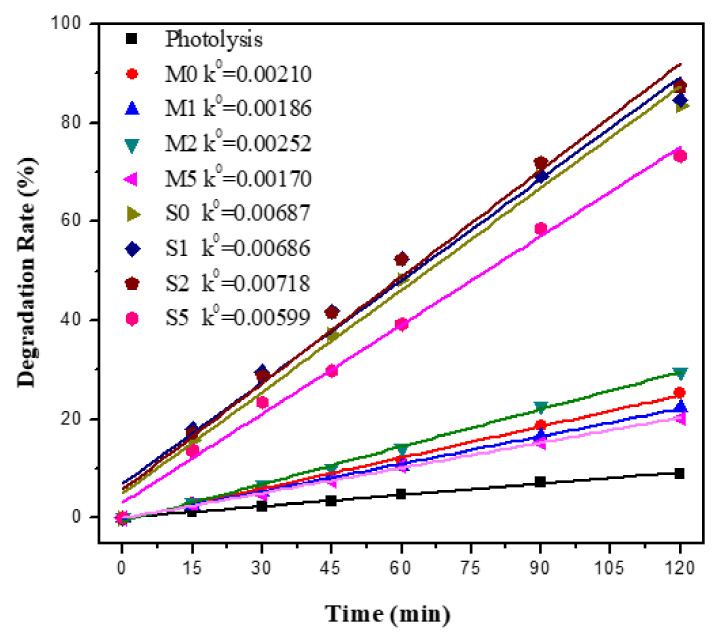
Degradation rate of MO and photocatalytic reaction rate of photocatalytic mortar blocks under UV light.

**Figure 9 materials-15-04031-f009:**
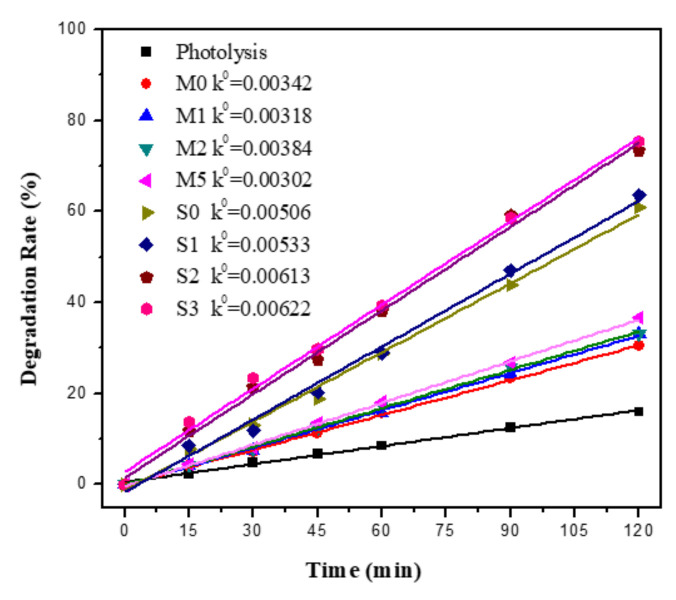
Degradation rate of methylene blue and photocatalytic reaction rate of photocatalytic mortar blocks under solar light.

**Figure 10 materials-15-04031-f010:**
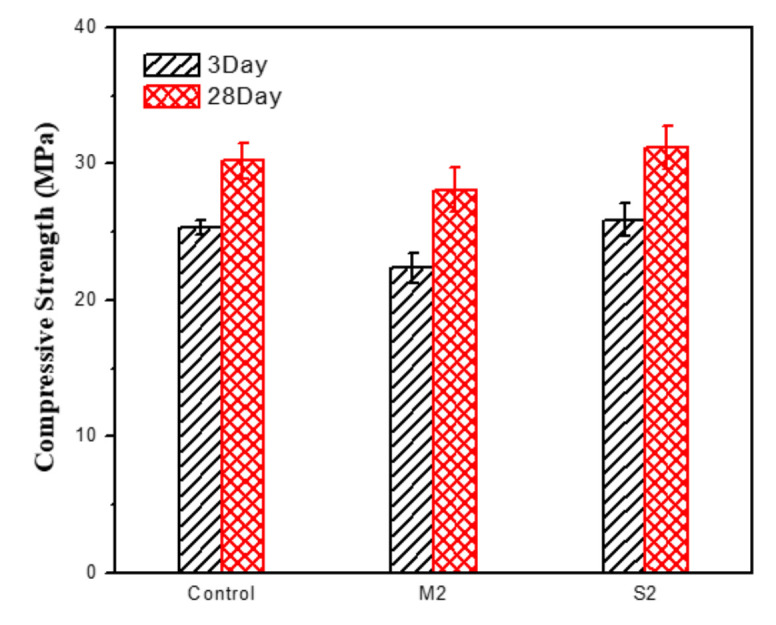
Compressive strength of Control, M2 and S2 at 7 days and 28 days.

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
