# Peer review of "Study on Photocatalytic Performance of Ag/TiO2 Modified Cement Mortar"

_materials, 2022, doi:10.3390/ma15114031_

Round 1
Reviewer 1 Report
1) Authors should indicate the lattice planes of corresponding diffraction peaks in figure 2.
2) How did you determine the particle size (20-50 nm) of synthesized materials?
3) It is recommended to report methylene blue degradation under UV light irradiation and methyl orange degradation under solar light irradiation by using photocatalytic cement mortar material.
4) Rephrase the references 1,4,7,8,9,16,19,22,25,28,31,35,39,40, 46 and 47 as per the journal format.
Reviewer 2 Report
In the present paper, the authors investigate the photocatalytic ability of Ag / TiO2 modified cement mortar to degrade methyl orange. The study needs to be improved before final acceptance. My comments are provided below:
- In the abstract section, information should be provided about the optimal process conditions, kinetic results and the effect of parameters.
- Keywords should be sorted alphabetically.
- In the abstract section, there is no information about color pollutants, methyl orange dye, various methods for removing color pollutants, dangers of color pollutants to the environment. It is necessary to add the mentioned contents in the introduction section. It is recommended to use the following articles:
- doi.org/10.1016/j.matchemphys.2022.126088.,doi.org/10.1016/j.chemosphere.2021.131632.,doi.org/10.1016/j.ijbiomac.2021.08.144.,doi.org/10.1016/j.chemosphere.2020.129419., doi.org/10.3390/molecules26082241.
-What is the reason for using cement mortar as a catalytic base?
- The relationship mentioned in line 304 must be corrected.
- Conditions for the demolition process must be provided below each of the figures.
- In Figure 6a compared to Figure 7a, the returns are different. If the reaction conditions are the same, how do you justify this contradiction?
- It is necessary to compare the ability and performance of the catalyst with other catalysts.
-Reuse of the catalyst should be considered.
- FTIR, BET analysis for the desired catalyst should be performed.
- A proposed mechanism for the process should be provided.
- The pHzpc value for the catalyst should be provided and the effect of pH on the efficiency of the degradation process should be presented.
Reviewer 3 Report
This article is interesting and can be published but few things should be well addressed be publiction
1. It is not clear about the phase of TiO2 because TiO2 is found more than one phase.
2. Density functional theory study will be helpful and strengthen these findings.
3. Diagrams needs better resolution like fig.4.
Reviewer 4 Report
1. Photocatalytic activity of Ag-TiO2 is already reported but there are few studies on the application of Ag-TiO2 in cement mortar, the author should show the state of the art materials (in Table) for understanding the current state.
2. What was the band-gap of different Ag modified TiO2 (1-5 %).
3. TEM is required to locate the Ag NPs in TiO2 and their structural investigation, size etc.
4. Author should list similar materials to show the comparison for the photocatalytic activity of Ag-TiO2.
5. Author should show the stability of the Ag-TiO2 towards photocatalytic activity ( used catalyst).
6. Introduction and discussion can be improved and the previous work can be cited https://doi.org/10.1021/acs.jpcc.9b05105; https://doi.org/10.1016/j.impact.2021.100345; https://doi.org/10.1016/j.apsusc.2021.150337; https://doi.org/10.1016/j.ceramint.2021.02.065
